# Melanoma Cells from Different Patients Differ in Their Sensitivity to Alpha Radiation-Mediated Killing, Sensitivity Which Correlates with Cell Nuclei Area and Double Strand Breaks

**DOI:** 10.3390/cancers16223804

**Published:** 2024-11-12

**Authors:** Or I. Levy, Anat Altaras, Lior Binyamini, Orit Sagi-Assif, Sivan Izraely, Tomer Cooks, Oren Kobiler, Motti Gerlic, Itzhak Kelson, Isaac P. Witz, Yona Keisari

**Affiliations:** 1Department of Clinical Microbiology and Immunology, Faculty of Medical and Health Sciences, Tel Aviv University, Tel Aviv 6997801, Israel; orlevy2@mail.tau.ac.il (O.I.L.); kaplunsky@mail.tau.ac.il (A.A.); liorbin111@gmail.com (L.B.); okobiler@post.tau.ac.il (O.K.); mgerlic@tauex.tau.ac.il (M.G.); 2The Shmunis School of Biomedicine and Cancer Research, The George S. Wise Faculty of Life Science, Tel Aviv University, Tel Aviv 6997801, Israel; oritsa@tauex.tau.ac.il (O.S.-A.); si0881@gmail.com (S.I.); isaacw@tauex.tau.ac.il (I.P.W.); 3The Shraga Segal Department of Microbiology, Immunology, and Genetics, Ben-Gurion University of the Negev, Beer-Sheva 8410501, Israel; cooks@bgu.ac.il; 4Sackler Faculty of Exact Sciences, School of Physics and Astronomy, Tel Aviv University, Tel Aviv 6997801, Israel; kelson@tauex.tau.ac.il

**Keywords:** alpha radiation, melanoma, squamous cell carcinoma, DNA damage, cell death, cell cycle arrest, nucleus size

## Abstract

This study investigated for the first time the ability of alpha particles to kill melanoma cells from different patients. it was evident that alpha particles can kill melanoma cells with inter-tumor heterogeneity. The cytotoxic effect of alpha radiation was stronger compared to the cytotoxic effect of photon radiation. The cytotoxic effect of alpha radiation was correlated with the formation of DNA double-strand breaks and cell proliferation arrest. Melanoma cells were more resistant to alpha radiation when compared with the response of another skin tumor, squamous cell carcinoma. The results are opening the way for alpha radiation treatment of melanoma with Diffusing alpha Radiation Therapy (Alpha DaRT).

## 1. Introduction

Cancer is a leading cause of death worldwide, accounting for nearly 10 million deaths in 2020, or nearly one in six deaths [1]. Melanoma remains a highly malignant type of skin cancer. While less common than basal cell carcinoma (BCC) or squamous cell carcinoma (SCC), melanoma is more dangerous because of its ability to metastasize to other organs more rapidly if it is not treated early [2]. Furthermore, advanced cutaneous melanoma tends to metastasize to the brain frequently with dismal prognosis [3].

Radiotherapy is used to treat benign and malignant diseases, and it can be used separately or in combination with immunotherapy, chemotherapy, and surgery. Therapeutic radiation is mostly delivered as low-linear energy transfer (LET) photon-based radiation termed External Beam Radiation Therapy (EBRT). EBRT therapy is an important treatment for cancer patients, with 50% of patients receiving radiation therapy during their illness, and it is considered to contribute to 40% of the cure of cancer [4]. External radiation may also use high-LET particle-based radiation. The use of high-energy protons or heavy ions from different sources (e.g., radionuclides) to the malignant area can cause more damage than photon-based radiotherapy [5]. Radiotherapy can also be delivered by placing radioactive sources near or inside the tumor, known as brachytherapy, which allows a higher total dose of radiation than the external beam treatment [6]. One of the advantages of brachytherapy as compared to EBRT in treating cancer patients is delivering an ablative radiation dose over a short period of time directly to the tumor tissue while sparing the adjacent organs. This method gave brachytherapy the status of standard therapy for a broad spectrum of malignancies [7].

High-LET alpha-particles possessing a LET of about 100 keV/µm can cause either excitation or ionization in biological matter, and because of their size, the track of the particle is almost linear, with a path length of several cell diameters [8]. High-LET radiations are known to have greater biological effectiveness per unit dose than low-LET ones in a wide variety of biological effects [9]. The signature of high-LET radiations, such as alpha radiation, is the formation of complex DNA damage that comprises closely spaced DNA lesions forming a cluster of DNA damage (double-strand breaks (DSBs) and non-DSB oxidative clustered DNA lesions (OCDL)) [10,11,12,13,14,15]. Highly destructive high-LET alpha particles have a short range in tissue (typically 30–90 μm for 5000–8000 keV alpha) [16] and are independent of dose rate, cell cycle during irradiation, and oxygen effects [17]. Thus, a single alpha particle hit to the nucleus has a 20–40% probability of killing the cell, and only a few hits are required to ensure cell lethality, mostly due to double-strand breaks caused to the DNA molecule of the cell by the particle [18,19,20,21,22]. Soyland and Hassfjell [23] provided evidence that particle hits in the cytoplasm did not significantly affect cell survival.

Alpha particle radiation, although highly destructive for tumor cells, is not used in EBRT due to its short range in tissue and is currently used as a radiopharmaceutical such as Xofigo (Ra-223).

Our laboratories developed an innovative intra-tumoral radiation (brachytherapy) tumor ablation method known as Diffusing alpha-emitters Radiation Therapy (Alpha DaRT). The principle of this technology is to introduce into the tumor an array of implantable metal sources that are coated with a low activity of radium-224 (3.7 days half-life). These sources emit Ra-224 daughter atoms into the tumor by recoil that spread inside the tumor by diffusion and convection (vascular and interstitial) over several millimeters around it [24,25,26]. These atoms release alpha particles, causing intensive tumor tissue damage (Figure 1).

The extent of radioactive atoms spread in the tumor may be affected by several characteristics, such as the tumor tissue compactness, disposal of radioactive atoms by the vasculature in the tumor, and cells’ sensitivity to radiation. There are several beneficial aspects of using alpha particle radiation as treatment against tumor cells: the short range that alpha particles travel in tissue is <100 µm, which keeps most of the radiation inside the tumor, reducing the risk to surroundings tissues and the treating staff; alpha particles can cause mostly double-strand breaks in the DNA in oxygen-low environments; studies have shown that only a few hits to the nucleus are required for reproductive cell death [27]. This treatment modality (alpha DaRT) has already been used for the treatment of patients with squamous cell carcinoma with excellent results [28,29].

Melanoma is the most common cause of skin-cancer-related death [30]. The 5-year melanoma-specific survival ranges from as high as 99% in patients with stage I–II disease to less than 10% for some patients with metastatic spread (stage IV) disease. Many melanoma patients carry regional and distant metastases at diagnosis. While surgery remains the main treatment for early, localized melanoma, stage III and IV melanoma require additional systemic therapies. The prognosis of patients was improved by targeted combination regimens of BRAF and mitogen-activated protein kinase (MEK) inhibitors and immune checkpoint blockade (ICB) and immune checkpoint inhibitors (anti-CTLA-4 and anti-PD1 antibodies). The role of adjuvant RT for high-risk resected melanoma remains limited [31,32].

Indications of radiotherapy are currently reduced, since melanoma is traditionally considered relatively resistant to conventional photon radiotherapy [33,34].

The major aim of this study was to investigate the cytotoxic effect of alpha particle radiation on four human-derived melanoma cell lines in vitro and compare and analyze several cell parameters which might affect sensitivity/resistance to alpha radiation. We also investigated how melanoma-derived cells compare to another skin-derived tumor, i.e., squamous cell carcinoma (SCC)-derived cells.

To the best of our knowledge, similar studies of this kind assessing the biological effect of alpha radiation on a battery of human melanoma have not yet been reported in the literature.

### Study Aims

Investigate the direct effect of alpha particle radiation on four cutaneous lines of human-derived melanoma and their brain metastatic derivatives.Look for variations among the variants’ cytotoxic effect.Compare the response of melanoma cells to alpha radiation to that of squamous cell carcinoma cells.Investigate the correlation between the cytotoxic effect of alpha radiation and the formation of double-strand breaks.Investigate the correlation between the cytotoxic effect of alpha radiation and the average size of cell nuclei.Compare the sensitivity of melanoma cells to alpha radiation with their sensitivity to photon radiation (gamma radiation).Investigate DNA double-strand breaks inflicted by alpha radiation in combination with modifiers of DNA repair mechanisms, such as ATR inhibitors (berzosertib) and topoisomerase 1 inhibitor (irinotecan), to enhance cell mortality.Investigate the cytotoxic effect of alpha radiation on synchronized cell culture after serum starvation to determine the significance of the cell cycle.

## 2. Materials and Methods

### 2.1. Cell Lines

We used four cutaneous human-derived melanoma cells—YDFR.C, DP.C, M12.C, and M16.C (C—for cutaneous) and their brain metastatic derivatives—which were obtained from murine models (CB3/CB2—for brain metastasis).

The development, culturing, and maintenance of human cutaneous melanoma variants YDFR.C and DP.C and human MBM variants YDFR.CB3 and DP.CB2 were previously described [35].

YDFR.C parental cells were kindly provided by Prof. Michael Micksche (Department of Applied and Experimental Oncology, Vienna University, Austria). DP-0574-Me (DP), UCLA-SO-M12 (M12), and UCLA-SO-M16 (M16) were kindly provided by Dr. Dave S.B. Hoon (Department of Translational Molecular Medicine, John Wayne Cancer Institute, Saint John’s Health Center Providence Health Systems, Santa Monica, CA, USA).

The cells were grown in RPMI-1640 medium (Gibbco, Grand Island, NY, USA) supplemented with 10% heat-inactivated fetal calf serum (FCS), 2 mmol/mL l-glutamine, 0.01 M HEPES buffered saline, 100 units/mL penicillin, 0.1 mg/mL streptomycin, and 12.5 units/mL nystatin. All cells were routinely cultured in humidified air with 5% CO_2_ at 37 °C.

Squamous cell carcinoma of the tongue, CAL 27 cell line, was used (https://www.atcc.org/products/crl-2095, accessed on 11 October 2023).

The identity of all cell lines used was authenticated using STR.

### 2.2. Cell Growth Curves

To determine the rate of division of our cell lines, melanoma cells were seeded in 96-well plates for several periods of incubation. Each well contained 7000 cells at the beginning. After each period of incubation, cells were fixed with 100% methanol and stained with Hemacolor reagents. Color was then extracted with 0.5% SDS solution and OD measurements were obtained in an Emax plate reader [36].

The division rate of YDFR.C, DP.C, and M12.C was approximately 48 h and that of M16.C was approximately 24 h.

### 2.3. In Vitro Irradiation with Alpha Particles Using an ^241^Americium (Am-241) Irradiator

#### 2.3.1. In Vitro Irradiation Apparatus

The in vitro irradiation setup (drawn schematically in Figure 2) consists of an alpha particle source containing Am-241 (half-life of 432.2 years) in equilibrium with its radioactive daughters (3.8-μCi (140 kBq)). The Am-241 source, due to partial screening, has an alpha emission rate of 50 ± 2.5 kHz. The source irradiates cells seeded on a 7.5 μm thick Kapton foil (polyamide film PRN-IF70, Pornat, Israel), which is thin enough to allow the alpha particles emitted from the source to reach the cells on its other side. The foil is held by an O-ring between two cylindrical stainless-steel parts, forming a well with an inner diameter of 9 mm in which the cells are seeded in culture medium as previously described [37]. During irradiation, the assembled well is positioned above the alpha particle source, so that the vertical distance between the source and the Kapton foil is 9.8 mm. Thus, the cells are exposed to a controlled flux of alpha particles, at an average dose rate of 0.25 Gy/min. The dose rate was estimated by measuring the spectrum and emission rate of alpha particles from the Am-241 source with an alpha-particle spectrometry system, supplemented by a Monte Carlo simulation of the full setup, using a dedicated MATLAB script utilizing SRIM 2013 [38]. The calculated mean alpha-particle energy, when passing through the cell layer, was 2.7 MeV, with an LET of 130 ± 3 keV/μm (where the variations reflect the radial position across the foil).

#### 2.3.2. Alpha Radiation in Vitro Procedure

Four cutaneous human-derived melanoma lines and their brain metastatic derivatives were grown in cell culture dishes (Corning, Corning, NY, USA) in RPMI 1640 Medium (Gibbco, Grand Island, NY, USA), containing 5% FBS, 1% PSN, and 1% l-Glutamine, until reaching sufficient confluency for seeding in the radiation cell holders.

Cells were harvested with 2 mL trypsin, incubated for 3 min, and then neutralized with 2 mL medium. Cells were centrifuged at 1200 RPM for 5 min; the supernatant was aspirated, and the pellet was mixed with 3 mL medium. Cell concentration was adjusted to 60,000 cells/mL. Radiation cell holders were assembled with Kapton film (irradiation cell holder) washed with ethanol 70% and left to dry out.

Cells were seeded on Kapton in the radiation cell holders, 300 μL per cell holder (18,000 cells per 300 μL), and incubated for 48 h. After 48 h of incubation, cells were irradiated and harvested for the viability assay procedure. Cells were seeded in 24-well plates and left for 5 days of incubation. Each experiment was divided into 5 groups of radiation periods (2 wells per radiation group) as follows: (1) 0 Gy (Control), (2) 0.35 Gy (3.45 min), (3) 0.7 Gy (7.5 min), (4) 1.4 Gy (15 min), (5) 2.8 Gy (30 min).

The same procedure was conducted for the CAL 27 cell line, with an exemption of the culture medium. Cells were cultured with DMEM medium (Gibbco) containing 5% FBS, 1% PSN.

### 2.4. Colorimetric (HemaColor) Viability Assay

The human melanoma cell variants do not form colonies after seeding in plates, and to test cell viability and multiplication after radiation, we used the HemaColor colorimetric assay [35]. After irradiation, 10,000 cells were harvested from the Kapton, centrifuged, counted, seeded in 24-well plates, and incubated for up to 5 days. When non-irradiated cells (control) reached 90–100% confluency, incubation was stopped and cell monolayers in all the wells were stained with HemaColor reagents (Sigma-Aldrich, Darmstadt, Germany) as described. After staining, plates were rinsed with tap water three times and filled again with water for decolorization for 5 min. Plates were then left to dry and the color was extracted with 0.7 mL/well of color extraction solution containing 0.5% sodium dodecyl sulfate (SDS) in double-distilled water (DDW) for 90 min. Three aliquots of 200 µL were removed from each 24-well plate and added to a 96-well plate for each group of treatment and for each cell line, respectively. Plates were read at 595 nm wavelength in an Emax spectrophotometer reader. The colorimetric assay was repeated 3–5 times for each cell line.

### 2.5. Exposure of Melanoma Cells to γ-Radiation

Melanoma cells were seeded in 24-well plates (20,000 cells/0.5 mL per well). Cells were left for 24 h to adhere. After 24 h, cells were irradiated with gamma radiation (BIOBEAM Gamma Irradiator provided by TAU SCIF) with doses of 0, 0.3, 0.7, 1.4, 2.8, 4.2, and 5.6 Gy and left for incubation until the control plate (0 Gy) reached full confluency. Cells then were stained with HemaColor reagents and photographed with a light microscope. Furthermore, color was extracted as described and triplicates of 200 µL samples for each cell line per radiation dose were transferred into 96-well plates and optical density was read with an Emax spectrophotometer with a wavelength of 595 nm. Data were collected and analyzed manually with Excel. Alpha radiation viability percentage data for our four cell lines were compared to the extracted data for gamma radiation.

### 2.6. Cell Nuclei Analysis

Four melanoma cell lines and CAL 27 were incubated in 96-well plates for 24 h. After 24 h, medium was removed, and cells were fixed with 4% Paraformaldehyde for 10 min. After fixation, wells were washed with PBS once and incubated with DAPI (Sigma, Darmstadt, Germany) for 5 min (dilution of 1:1000). After incubation with DAPI, wells were washed once with PBS and left to dry in the dark. Images were taken with a Nikon Eclipse TI fluorescent microscope with a scale bar of 100 µm ratio. Images were then analyzed with ImageJ/FIJI (ImageJ open-source software 2.16.0) for measurement of the cell nucleus in each of our cell lines.

Average perimeter of the nucleus along with the standard deviation, cell radius, diameter, and area were calculated manually with Excel. Comparisons of cutaneous cell lines were presented in radar plots to compare between cells. One-way analysis of variance (ANOVA) test was conducted along with Tukey’s Honestly Significant Difference (HSD) post-hoc analysis on the derived perimeter of each cell variant.

### 2.7. Double Strand DNA Breaks Signaling

A distinctive marker of the effect of alpha particle radiation is consensually agreed to be double-strand breaks (DSB) in the DNA. This breakage in the DNA activates DNA repair pathways. The phosphorylation of the histone gamma-H2Ax (γ-H2aX) and its binding to the DNA is an indicator of the damage to the nucleus [39]. It has been stated that radioresistant cancers may have acquired an increased proliferative capacity and an upregulation in DNA repair pathways [40]. To determine the degree of radio resistance/sensitivity in radiated cells, γ-H2aX foci may serve as a convenient comparative marker and can be visualized by an immunofluorescence method.

For the immunofluorescence assay, primary Anti-phospho-Histone H2A.X (Ser139) antibody, clone JBW301, was purchased from Merck (Rehovot, Israel). The secondary antibody, Goat Anti-Mouse IgG H&L (Alexa Fluor^®^ Abcam 488), was purchased from Zotal Inc. (Tel Aviv, Israel), and DAPI reagent was purchased from Sigma. Pictures were taken using a Nikon Eclipse TI fluorescent microscope (Optical Industries, Tokyo, Japan).

Four human-derived melanoma cell lines and CAL 27 cells were seeded on Kapton in the irradiation apparatus at a concentration of 40,000 cells/0.5 mL in each radiation cell holder and left to adhere for 24 h. After 24 h, cells were irradiated with 1.4 Gy of alpha radiation (15 min irradiation period). Culture medium was removed, and cells were fixed with 4% para formaldehyde in PBS (Sigma) for 12 min. After fixation, a circular area of kapton, with fixed cells, was cut with a needle and placed on stretched parafilm paper in 24 well plates. After fixation, cells were washed with PBS three times for 5 min each. Cells were then permeabilized with 0.5% triton -X solution in PBS for 10 min and washed twice in PBS. For blocking, cells were treated with 10% FBS in PBS solution for 2 h. Primary antibody diluted 1:250 in blocking solution was added to the cells and left overnight at 4 °C. After the primary antibody was removed, cells were washed with PBS three times for five minutes each. Secondary antibody diluted 1:200 in PBS was added to the cells and left for 0.5 h at RT in the dark. After the secondary antibody was removed, cells were washed once with PBS for 5 min, and DAPI diluted 1:1000 in DDW was added. After the DAPI was removed, cells were rinsed with PBS and the circular foils were mounted on glass slides and coverslip and left to dry overnight. Photos were taken and foci were counted manually with ImageJ.

A one-way analysis of variance (ANOVA) test was conducted along with Tukey HSD post hoc analysis.

### 2.8. Viability Assay: D50 Measurements

The D50 calculation is the measure of the radiation dose needed for 50% cytotoxicity/viability for each cell variant. The D50 value was used to compare the relative sensitivity to radiation of the different cells.

The value of D50 was calculated with the linear line formula that was given by the manual calculations of the converted viability percentage.

As data were extracted from the spectrophotometric measurements of remaining cells after radiation, a one-way analysis of variance (ANOVA) test was conducted along with Fisher’s Least Significant Differences (LSD) post hoc analysis (α = 0.05). To compare our variants statistically, we defined the dependent variable in the ANOVA test as D50.

### 2.9. Serum Starvation and Alpha Radiation

Chromatin structure may affect sensitivity to photon radiation [41], and it was suggested that synchronized cell culture induced by serum starvation can induce chromatin condensation in the G0/G1 phase of the cell cycle [42]. Thus, our intent was to investigate the effect of alpha radiation after serum starvation in a synchronized cell culture.

To determine if melanoma cell lines can undergo cell cycle arrest upon serum starvation, cells were seeded in wells, at 100,000 cells per well, with RPMI-1640 medium containing 10% FBS serum, 1% PSN solution, and 1% L-glutamine for 24 h. After 24 h, medium was replaced with RPMI-1640 medium (Gibbco) containing 0.5% FBS (Gibbco) serum (starvation medium) for 24 h and 48 h. After incubation in starvation serum medium, cells were harvested into 15 mL tubes and centrifuged for 5 min at 1200 RPM. Cells were then fixed with 70% ethanol and left at −20 °C for 24 h. Cells then were centrifuged in full medium and collected into FACS tubes in 300–400 mL of PBS (Sigma). Cells were stained with 5 mL of propidium iodide (Sigma) and analyzed in an S1000EXi FACS flow cytometer (Statedigm, Inc., San Jose, CA, USA).

### 2.10. Treatment of Melanoma Cells with Berzosertib or Irinotecan and Alpha Radiation

Melanoma cells were incubated until they reached sufficient confluency. Cells were seeded in 96-well plates at a concentration of 25,000 cells/mL, each well containing 5000 cells/200 µL, and left to adhere for 24. After 24 h of incubation, culture medium was removed and berzosertib or irinotecan were added with increasing concentrations of the drug and further incubated for 72 h.

After incubation, the drug was removed, and plates were stained with HemaColor reagents. Plates were photographed and color was extracted with 0.5% SDS solution in DDW. Plates were read with an Emax spectrophotometer, and raw data were obtained and calculated manually with Excel.

Combination treatment of the drugs and alpha radiation was performed with cells seeded in radiation holders upon Kapton (20,000 cells per 300 µL for each Kapton). Cells were left to adhere for 24 h. After 24 h, Irinotecan or Berzosertib at sublethal concentrations were respectively added and left for another 24 h incubation. Cells were then irradiated, harvested, and seeded in 24-well plates at a concentration of 10,000 cells per well and left for incubation until control wells were confluent (approximately 5 days).

### 2.11. Cell Growth Kinetics Analysis

Melanoma cells were seeded in radiation cell holders at a concentration of 100,000 cells per ml (50,000 cells per 500 µL per radiation holder) for 24 h to adhere and then irradiated with increasing doses of alpha radiation (0–5.6 Gy). Cells were removed from the radiation cell holder and seeded in 96-well plates, with triplicates of 10,000 cells per well for each radiation dose. Cell growth and cell death were measured with an ~IncuCyte 2022B Rev2~ Incucyte^®^ SX5 machine (Ann harbor, Detroit, MI, USA) for 5 days with propidium iodide (PI). Analysis of total cell count and PI positive cells was conducted with ~2022B Rev2~ Incucye2022BRev2 software (Ann harbor, Detroit, MI, USA) AI analyses.

## 3. Results

### 3.1. Sensitivity of Melanoma Cell Lines and Their Brain Metastatic Variants to Alpha Radiation

Alpha particle radiation potential as a cytotoxic agent against human-derived tumors has been discussed in the literature and reported in previous works from our laboratory [27,37,43,44,45]. In this study, we investigated the cytotoxic effect of alpha radiation on human-derived melanoma cell variants in vitro. It has been stated that melanoma is a type of skin cancer that is relatively resistant to conventional radiotherapy (photon radiation) in vitro [46].

Four cell lines of human-derived cutaneous melanoma and their brain metastatic derivatives were irradiated with alpha-particle fluxes in vitro (Figure 2). The exposure to ongoing doses of alpha radiation revealed that our cell variants were sensitive to the cytotoxic effect of alpha radiation. HemaColor viability assays showed 54–71% cytotoxicity at the highest dose (2.8 Gy) of alpha radiation. Comparison with the sufficient dose for killing 50% of cells (D50) using an analysis of variance (ANOVA) test showed an inter-variant variance among our melanoma cell lines. Furthermore, Tukey HSD post hoc analysis showed significant inter-variant variance between the cell lines DP.C (the most sensitive, D50 = 1.51 ± 0.51 Gy) and M16.C (the most resistant, D50 = 2.64 ± 0.74 Gy) and non-significant yet considerable variance between the cell lines YDFR.C (D50 = 2.05 ± 0.64 Gy) and DP.C to M12.C (D50 = 1.87 ± 0.54 Gy). The brain metastatic variants showed sensitivity very similar to their skin parent cell (Table 1). According to the post hoc analysis, we continued our research on the cutaneous cell lines, trying to reveal properties of cell biology that may be correlated with this variance.

### 3.2. Comparison of Melanoma Cells with Squamous Cell Carcinoma Cells

The results indicated a variable response to alpha radiation differing between the cell lines. It was interesting to compare the response of the melanoma cells to another skin-derived tumor such as squamous cell carcinoma, and the CAL 27 cell line (SCC of the tongue) was used. CAL 27 cells were irradiated and tested with the same procedure of cell viability assays. CAL 27 (D50 = 1.37 ± 0.07 Gy) was more sensitive than all melanoma cells, but paired *t*-test analysis, which was conducted on CAL 27 and the cutaneous cell lines, revealed significant variance between CAL 27 and the most resistant cell, M16.C (*p* = 0.0264) (Table 1 and Table 2). Viability curves also emphasized the higher sensitivity of CAL 27 cells compared to cutaneous melanoma (Figure 3).

### 3.3. Morphology of the Cell Nucleus as a Determinant of the Cytotoxic Effect of Alpha Particle Radiation

Cells exposed to alpha particles are subjected to a stochastic number of alpha particle traversals through their nuclei. Cell survival studies suggested that only a few hits to the nucleus are required to induce a lethal effect. Moreover, alpha particles that hit the cytoplasm do not significantly affect cell survival [23,47]. Different cell types are characterized by different nuclear morphologies in general and nuclear areas in particular, and it was previously shown by Lazarov et al. [27] that there is a correlation between the number of hits by alpha particles and the nuclear area of cells. Thus, the nuclear morphology of human-derived melanoma cell variants was assessed and related to the differences in the cytotoxic effect of alpha radiation on these cells and on SCC.

The nuclear area of the cell lines was determined by immunohistochemistry assay where nuclei were stained with DAPI. After staining with DAPI, photos were taken with a Nikon eclipse TI fluorescent microscope, with a scale bar of 100 µM, and we analyzed the photos with ImageJ/FIJI software (Figure 4).

As seen in Table 3, the mean perimeter of each cell line was calculated with the standard deviation and an ANOVA test was conducted. Results showed a statistically significant difference between all four cell lines (*p* = 2 × 10^−26^), and post hoc analysis revealed strong significant differences between the pairs YDFR.C:M12.C (*p* < 0.0001), YDFR.C:M16.C (*p* < 0.0001), DP.C:M12.C (*p* < 0.0001), and DP.C:M16.C (*p* < 0.0001). No significance was shown between YDFR.C:DP.C (*p* = 0.33) or M12.C:M16.C (*p* = 0.85). Other properties of nuclear morphology were calculated according to the derived radius, and multi-parametric evaluation of the cell nucleus properties among pairs of cell variants are shown in spider web graphs. According to the findings, the most sensitive cell (DP.C) had a larger nucleus compared to the least sensitive cell (M16.C) (Figure 4). Nuclear area and morphology may be a macroscopic cause for the variance in cytotoxicity of alpha radiation on human-derived cutaneous melanoma cell variants (Figure 5).

It is important to note that the squamous cell line, CAL 27, which was more sensitive than the most sensitive melanoma cell, DP.C, has a much smaller nucleus size, in the range of the more resistant melanoma cell M16.C (Table 3).

### 3.4. Exposure of Cells to Alpha Particle Caused DNA Double Strand Breaks (DSB) as Measured by γ-H2aX Foci

DSB in the DNA are one of the main causes of cell death after exposure to ionizing radiation [48]. Thus, we questioned whether more sensitive cells have more double-strand breaks when compared to the more resistant cell lines.

Cutaneous melanoma cell lines and CAL 27 were irradiated with a dose of 1.4 Gy of alpha radiation and stained with the anti-phosphorylated histone γ-H2AX antibody (Figure 6), and foci per cell were counted and analyzed using ImageJ/FIJI software. The numbers of DSB were 7.68 ± 2.12 for YDFR.C, 9.61 ± 2.3 for DP.C, 6.17 ± 2.23 for M12.C, and 6.9 ± 1.65 for M16.C. ANOVA test for foci per cell per cell line with a confidence interval of 95% showed significant variance between our cells (*p* = 1.06 × 10^−8^). Post hoc analysis of the results revealed significant variance among pairs of cell variants: DP.C:M12.C (*p* = 1 × 10^−8^), DP.C:M16.C (*p* = 5.64 × 10^−6^), DP.C:YDFR.C (*p* = 0.002), M12.C:YDFR.C (*p* = 0.037). Yet non-significant differences were also obtained in the analysis: M12.C:M16.C (*p* = 0.53) and M16.C:YDFR.C (*p* = 0.49). Concluding the analysis results, the number of DSB in the DNA vary among human-derived cutaneous melanoma, and a higher number of DSB was observed in the more sensitive cell lines. It is interesting to note that the number of DSB in the CAL 27 (SCC) cell line (6.67 ± 1.38) was low compared to the melanoma cells and in the range of the more resistant melanoma cells.

To compare and correlate the sensitivity to alpha radiation, the cell nuclei size, and the DSB numbers in the melanoma cells, we include a summary table (Table 2). There is a direct correlation between sensitivity to radiation (D50), number of DSB, and the size of the nucleus for the cell lines DP.C, YDFR.C, and M16.C. The cells with larger average nuclei sizes were observed to have more DSB per cell and were more sensitive to alpha particles (lower radioactive dose needed to kill them). The exception was cell line M12.C, which had the smallest cell size and the lowest number of DSB per cell but was not the most resistant cell line. The comparison of melanoma cells to the SCC cell line CAL 27 showed that this cell line had the smallest nucleus and low DSB but was still the most sensitive cell.

### 3.5. Alpha Radiation Is More Cytotoxic to Melanoma Cells in Comparison with Gamma Radiation

In a previous study, it was observed that alpha radiation is more lethal for human SCC (FaDU) and pancreatic carcinoma (Panc01) cells than X-rays [27]. Thus, it is important to make this comparison for the four human melanoma cell lines. The cutaneous cell lines of human-derived melanoma were seeded in 24-well plates (20,000 cells/0.5 mL per well) and left for 24 h to adhere. After 24 h, cells were irradiated with gamma radiation (BIOBEAM 8000 Gamma Irradiator, Gamma Service Medical GmbH, Leipzig, Germany) with doses of 0, 0.3, 0.7, 1.4, 2.8, 4.2, and 5.6 Gy and further incubated until the cells in the control plate (0 Gy) reached full confluency. Cell viability was assessed with the Hemacolor viability assay.

Comparison of the cytotoxic effect of alpha radiation and gamma radiation (Figure 7) showed that alpha radiation was significantly more cytotoxic than gamma, comparing all four cell lines (*p* < 0.05). The use of higher radiation doses with gamma radiation highlighted alpha particle radiation’s efficiency in destroying the notorious relatively resistant melanoma.

It is interesting to note that the cell line most sensitive to alpha radiation (DP.C) was also more sensitive to photon radiation.

### 3.6. Cell Cycle Synchronization and Proliferation Arrest Did Not Affect Cell Killing by Alpha Radiation

In this study, serum starvation of cell cultures was used to induce cell cycle synchronization. Serum-starved cells can undergo a shift in cell cycle by arresting normal proliferation and entering the G0/G1 phase. It has been suggested in the literature that serum-starved cells can undergo chromatin condensation [42]. Thus, we checked if proliferation arrest and chromatin condensation may affect the cytotoxic effect of alpha particle on irradiated melanoma cells.

### 3.7. Cell Cycle Analysis

FACS analysis of the four cell lines revealed coherent cell cycle arrest in three of the four cell variants. The effect was most pronounced in the DP.C cell line, which exhibited the highest sensitivity to alpha radiation-mediated killing (Figure 8). As for the M12.C cell line, cell cycle analysis showed an elevation in the sub-G1 phase, which could imply contribution to the breakage of proliferation (N = 6).

### 3.8. Effect of Cell Starvation on the Killing of Cells by Alpha Radiation

To evaluate the effectiveness of serum starvation on the enhancement of alpha radiation cytotoxicity, we first measured the multiplication rate of our cell lines to determine when to stop the incubation after the combined treatment.

After the determination of each cell line’s growth rate, the effect of starvation on cell death inflicted by alpha radiation combination treatment was determined using DP.C cells, the most sensitive cutaneous melanoma. Cells were seeded on Kapton in irradiation holders in two groups: alpha (treated with full medium) and full starvation pre-radiation (S). Cells were then irradiated and seeded in 96-well plates and left for a time equal to one division cycle (24–48 h, depending on cell line) as described in Section 2.9.

Exposure of alpha radiation-sensitive starved cells to alpha particles non-significantly increased the level of cytotoxicity at the low level of radiation (0.7 Gy), from 12.5% to 29.65%, but not at higher radiation levels. This may indicate that lack of proliferation and probably chromatin condensation does not have a significant effect on alpha particle-inflicted cell damage and death.

### 3.9. Combination Treatment with DNA Damage Repair Modifiers and Alpha Radiation

This part of the study was dedicated to finding ways to enhance the cytotoxic effect of alpha radiation. Clinical approaches and the literature suggest a combination of chemotherapy with radiation as a multi-model approach for better killing of malignant cells.

In this study, we used two agents, the ATR inhibitor berzosertib, which interferes with the ability of the cells to detect DSB and thus promotes cell death, and the topoisomerase I inhibitor irinotecan, which eventually leads to the inhibition of both DNA replication and transcription.

The results show that all four cell lines are sensitive to berzosertib and irinotecan.

Berzosertib killed fifty percent of DP.C at 356 nM, M12.C at 727 nM, YDFR.C at 953 nM, and M16.C at 1012 nM. Irinotecan killed fifty percent of DP.C at 10.4 mM, M12.C at 13.3 mM, YDFR.C at 35.5 mM, and M16.C at 8.84 mM. It is interesting to note that there is a difference in the response of the cells to each agent. The sensitivity of the cell lines to berzosertib did not match the sensitivity to irinotecan and was also different from their sensitivity to alpha radiation.

Next, we examined if combination treatment with the drugs enhanced alpha radiation-mediated killing. Four groups were used: control (treated with full medium), and three concentrations each of berzosertib or irinotecan, according to the drug calibration curves stated earlier. Cells were then irradiated, harvested, seeded in 24-well plates, and left for incubation until control wells were confluent (approximately 5 days).

Combined treatment with berzosertib—The results of three repetitions of the colorimetric viability assay for each cell line were collected and summarized. The results showed that melanoma cell lines responded to the cytotoxic effect of the ATR inhibitor itself. Yet the cytotoxic effect of alpha radiation (0.7, 1.4, and 2.8 Gy) was partially, but non-significantly, enhanced by berzosertib at the range of 100, 300, and 500 nM.

Combined treatment of alpha radiation with irinotecan—Colorimetric viability assays with the YDFR.C cell line were performed and showed that low concentration of irinotecan (1 µM) significantly elevated the cytotoxic effect of 0.7 Gy alpha radiation from 20 to 40% (*p* = 0.0233), and at 1.4 Gy, cytotoxicity was increased from 38% to 55% (*p* = 0.0008). Photos of the treated cells are presented in Figure 9.

### 3.10. Cell Growth Kinetics Analysis

Our results so far showed that although alpha radiation caused cell death in melanoma cells, there was still a percentage of viable cells continuing to grow despite the aggressive nature of the alpha particles hitting their nuclei and breaking their DNA. Thus, we compared the cell death kinetics after radiation of the most sensitive cell, DP.C, to the most resistant one, M16.C, to acquire more knowledge about the cytotoxic effect of alpha radiation.

Melanoma cells were seeded in radiation cell holders and then irradiated with increasing doses of alpha radiation (0–5.6 Gy). Cells were removed from the radiation cell holder and seeded in 96-well plates. Cell growth and cell death were measured by an ~IncuCyte machine for 5 days with propidium iodide (PI), and analysis of total cell count and PI positive cells was performed.

Results presented in Figure 10 show the total cell count during the five days after radiation. It is evident that there is a difference between the behavior of the sensitive DP.C cells and the more resistant M16.C cells. Sensitive cells fail to recover even after exposure to a low radiation dose (0.7 Gy), while the resistant M16.C cells start to proliferate within one day, and recovery is evident even after the highest dose of 5.6 Gy.

## 4. Discussion

### 4.1. Cancer Radiotherapy

Radiotherapy is a powerful tool against cancer. The treatment of solid tumors with different approaches of radiation therapy is widely discussed in the literature [49]. Malignant melanoma is a type of skin cancer that is known for its relative resistance to photon radiation [33,34,46]. The need to overcome the radio resistance of melanoma directed our team to investigate the potential use of alpha particles radiation on human melanoma. Our laboratories developed an ablation method using alpha particle radiation (DaRT) as mentioned before. This technique, using alpha radiation, proved to induce effective tumor response in patients with squamous cell carcinoma (SCC) [28,29].

A single alpha particle hit to the nucleus has a 20–40% probability of killing the cell, and only a few hits are required to ensure cell lethality [9,10,11], mostly due to DNA double-strand breaks (DSBs) [10]. Densely ionizing radiations induce DSBs that are spatially correlated, more complex (i.e., associated with other lesions), and more difficult to repair than those induced by X- or gamma rays [9,10,50]. Using gene expression as an endpoint, it was also shown that stressful effects are transmittable from cells exposed to high-LET IR to non-irradiated cells [51].

These characteristics suggest alpha irradiation as a natural candidate for the treatment of cancerous tissues. Therefore, alpha particle radiation was selected to be researched for its cytotoxic effect on a panel of human-derived melanoma cell variants.

It should be mentioned that the melanoma cells used in this study were previously shown to exhibit inter-tumor heterogeneity. Employing xenograft models of human melanomas in nude mice, differential functional responses of these melanoma cells from different patients to GM-CSF were detected both in vitro as well as in vivo. Whereas cells of one melanoma acquired pro-metastatic features following exposure to GM-CSF, cells from another melanoma either did not respond or became less malignant [52].

In the current study, firstly, we measured the response of human melanoma cells to alpha radiation, secondly, we studied the variance in response within a group of cells from the same histotype, and thirdly, we looked for cell characteristics which might correlate with sensitivity/resistance of melanoma cells to alpha radiation.

Our study showed that in vitro exposure of human melanoma cells to alpha radiation killed the melanoma cells with variations in sensitivity. In vivo studies in melanoma-bearing mice treated with a peptide conjugated with an alpha-emitting radio-labeled nuclide, showed a decrease in tumor growth rate, extended mean survival rate of the subjects, and in many cases, complete remission of the disease [53].

The alpha radiation dose required for killing 50% of the cells (D50) was measured as a quantitative variable for the comparison of the cytotoxic effect of alpha radiation among all melanoma cell lines. Considerable variations in cytotoxicity levels inflicted by alpha radiation were observed among the four cutaneous cell lines. Post-hoc analysis after ANOVA of cutaneous cell lines revealed that the DP.C cell line was more sensitive than the other three (YDFR.C, M12.C, M16.C) and significantly more sensitive compared to the M16.C cell line. D50 measurements suggest that some cell lines inside the population of cutaneous melanoma needed almost a double dose of radiation to exhibit the same cytotoxicity as the others. In comparison, the melanoma cells showed high resistance to photon radiation with small variation among them, although the DP.C cell line was also the most sensitive to photon radiation. The variance showed with both radiations, and in particular the variance in cytotoxicity of alpha radiation, emphasized the heterogeneity of human melanoma response after radiation treatment and should be optimized if used clinically. A detailed study performed with many melanoma cells reported in vitro heterogeneity to photon radiation and the ability to overcome resistance to radiation by Braf inhibition [54].

As heterogeneity in response to alpha radiation was observed in our viability assays, we further looked for cellular properties which may contribute to the variance in response to alpha radiation among our cell lines. As alpha particles can penetrate the cell nucleus and break its DNA in different locations, the cell nucleus is determined as a target for these particles. Thus, we looked for a correlation between the nuclear area of the cells and their sensitivity to alpha radiation. Measurements of the cell nuclei, perimeter, and the derived radius were performed for each of our cutaneous cell lines. Variance was obtained between the YDFR.C and DP.C compared to M12.C and M16.C cell lines. Comparison of nuclear size and sensitivity to alpha radiation indicated that the most sensitive cell to alpha radiation had the largest nucleus (DP.C) while the resistant cell had a much smaller nucleus (M16.C) (Table 2). These findings suggest that when comparing our cutaneous melanoma cell lines, the larger the nucleus, the more sensitive the cell line to alpha radiation, with the exception of line M12.C.

This finding aligns with previous studies. Microdosimetric analysis of other cancer cell lines subjected to alpha radiation showed that the nuclear area of the cells irradiated with alpha radiation is a parameter to be considered when dealing with different tumor cell lines [27]. Furthermore, it has been well discussed that considering broad beam radiation using heavy ions, the number of hits is determined by the cell nucleus [55]. Moreover, a recent paper suggested that the cell and cell nucleus size may impact the biological effect of radiation therapy [56].

### 4.2. Alpha vs. Photon Radiation-Inflicted Damage in Melanoma-Derived Cells

In vitro examination of the irradiated melanoma cells in the present study showed a decrease in proliferation along the dose scale, and cytological staining with HemaColor regents revealed swollen membranes and broken nuclei along with well-observed apoptotic bodies. Furthermore, treatment with alpha radiation was much more efficient in killing the cells compared to irradiation with photon radiation (gamma radiation) (Figure 7). These results are corroborated by previous studies in which treatment of cells with alpha radiation or X-rays was compared [27,37]. Interactions of a-particle tracks with the cell may result in an all-or-nothing event, and if there is a hit, it is likely to be severe clustered damage that results in several small fragments, whereas with the sparser open structure of X-rays, a hit may be more likely but can result in just one or a few events, thus reducing the number of small fragments produced compared with alpha-particles [57].

Several studies have investigated DNA damage and repair mechanism differences between gamma and alpha radiation.

A direct comparison of the effect of irradiation of HeLa and oropharyngeal squamous cell carcinoma by high-LET α-particles and protons or by low-LET protons or X rays/γ-radiation revealed that the H2B ubiquitylated signaling and repair of complex DNA damage (CDD) is induced only by high LET, which contributes significantly to cell survival after irradiation [58]. In another study, thyroid follicular carcinoma cells were irradiated with 3 Gy of gamma (^60^Co source), (3) Neutrons (N) (neutron beam alone), or BNCT. The results showed that the number of nuclear γH2AX foci was higher in the gamma group than in the N and BNCT groups. Yet the focus size was significantly larger in BNCT compared to other groups. [59]. Haro and coworkers created stable myeloid leukemia HL60 cell clones radioresistant to either γ-rays or α-particles. Cross-resistance to each type of IR was observed, but resistance to clustered, complex α-particle damage was substantially lower than to equivalent doses of γ-rays. A more robust repair of DNA double-strand breaks was evident when damage was induced by γ-rays, compared to α-particle-inflicted damage. The resistant phenotype was driven by changes in apoptosis, late G2/M checkpoint accumulation that was indicative of increased genomic instability, and stronger dependence on homology-directed repair [60].

From the practical viewpoint, the results emphasize the possible role of alpha particles as a suitable treatment for hard-shelled tumors such as melanoma using a methodology such as alpha DaRT.

### 4.3. DNA Damage and Repair

To further elucidate the cause of the variations in cytotoxicity in our cells, we investigated the formation of DNA double-strand breaks (DSB) by alpha radiation.

Double-stranded breaks in the DNA are deadly to the cell nucleus, and accumulation of these can lead to cell death and, in some cases, to cell senescence [61]. It has been stated that alpha radiation, due to its higher LET, is more effective than other ionizing radiations such as gamma radiation in breaking DNA [62]. The use of the phosphorylated histone γ-H2ax marker to detect DSB in our irradiated cell lines nuclei revealed that all our cutaneous cell lines had phosphorylated foci (Figure 6, Table 3) immediately after uniform radiation with 1.4 Gy. Furthermore, analysis of the number of foci of each cell line confirmed that cells more sensitive to radiation had a larger nucleus (DP.C) and showed more phosphorylated foci than the other cells. Although there are other properties involved in the heterogeneity in response to alpha radiation, we can conclude that nuclear morphology and the mean number of DSB foci can be key properties in determining the variance among our cutaneous melanoma cell lines.

High-LET ionizing radiation has a higher RBE compared to low-LET radiation, probably because it forms clustered lesions which hamper the DNA repair machinery of the cell and not because of a greater overall number of DSBs [63,64].

Radiation-induced DNA-damage, especially DNA DSBs, triggers activation of the ATM-initiated signaling cascade to arrest cell division until repairs can be made. This cascade of DNA-damage responses is composed from the following core components: the signals, sensors of signals, transducers, and effectors [65,66]. The primary transducer of the DSB alarm is the nuclear protein kinase ataxia-telangiectasia mutated (ATM) [67]. Following ATM activation, the histone H2AX is phosphorylated (to become γ-H2AX), which in turn elicits a sequence of signaling events [68]. H2AX is one of the most genetically conserved H2A-variants and is present in chromatin at levels that vary between 2–25% of the H2A pool, depending on the cell line and tissue examined [69,70]. H2AX was revealed to be a key player in the cellular responses to DNA damage after the discovery that it is locally phosphorylated on a conserved serine (Ser139 in mammals), to generate γ-H2AX in the vicinity of DSBs, which is implicated in amplifying the DNA damage signals [68,71,72]. γ-H2AX foci formation is accepted as a marker of DSBs [73], and all three major PIKK (PI3K-like protein kinases) members, ATM, ATR, and DNA-PKcs, have the potential of phosphorylating H2AX [65,74,75,76,77].

### 4.4. Melanoma Cell Response to Alpha Radiation in Comparison with Squamous Cell Carcinoma

It was important to investigate whether the correlation between cell killing by alpha radiation, nuclear morphology, and the number of DSB which was found within the melanoma group of cells is also applicable for another skin tumor, squamous cell carcinoma. We compared our cutaneous melanoma cell lines to a SCC line (CAL 27), assessing its D50 measurement, nuclear area, and the mean phosphorylated foci after radiation. The SCC cell line, CAL 27, had a similar number of DSB (6.67) and nucleus size (175.49 μm^2^) to the more resistant melanoma cells. Yet it was more sensitive to alpha radiation than all melanoma cells. This might indicate that other factors such as DNA damage and repair mechanisms and death-related signals, to name a few, may also figure importantly in the sensitivity to alpha radiation.

An interesting finding by Lemaitre and coworkers on DSB and the repair pathways controlling them revealed that the nuclear position of the DSB dictates the choice of DNA repair pathways. It was suggested that there is a spatial regulation of the DNA damage response to DSB and that the position of the DSB may be a crucial variable in maintaining the response to the damage [78]. This means that DSB in different cell types may occur at different locations in the nucleus triggering mechanisms of repair or cell death.

Another important observation in our research relates to the death kinetics of the cells after alpha radiation. Live cell imaging showed, from the perspective of time, that the cutaneous melanoma cell lines had different recovery patterns after radiation which were radiation-dose-dependent (Figure 10). Each cell line had a different pattern of growth trend and different time that it started to recover and proliferate. For example, the radiation-sensitive cell line, DP.C, with the largest nuclear area and the most phosphorylated foci after radiation, hardly recovered at all radiation doses. In contrast, the M16.C cell line, the most resistant cell line, regained a higher proliferative capacity as compared to DP.C.

### 4.5. DNA Damage and Chromatin Configuration

Chromatin structure was suggested to affect the extent of DNA damage and repair after ionizing radiation. Although the effects of chromatin structure on the damage are not fully understood, several mechanisms have been proposed. Chromatin condensation may have a protective effect against radiation damage. Thus, it was shown that tightly packed chromatin (heterochromatin) is more resistant to DSB in the DNA after ionizing radiation and that dividing cells are more sensitive to photon radiation [41].

It was suggested that synchronized cell culture induced by serum starvation can induce chromatin condensation in the G0/G1 phase of the cell cycle [42]. Thus, our intent was to investigate the effect of alpha radiation after serum starvation in a synchronized cell culture.

Since the state of the DNA and chromatin at the time of radiation might be crucial for the radiation-inflicted damage and the function of DNA repair mechanisms, we attempted to resolve the issue of chromatin structure in relation to response to alpha particles. The DNA structure is classified into two states: condensed (heterochromatin, HC) and relaxed (euchromatin, EC) [79]. In condensed chromatin, high-LET radiation, such as alpha particles, might cause more damage than that in the regular relaxed state. This is primarily because alpha radiation does not depend on the oxygenated state of the target and secondly because few alpha particles may cause complex DNA breaks and higher lethal damage would be achieved, and condensed chromatin might be less favorably for the function of repair mechanisms. It was reported that the condensed state restricts DNA DSB repair and damage response signaling and that ATM signaling enhances heterochromatin relaxation in the DSB vicinity and that this is a prerequisite for HC-DSB repair [80]. Furthermore, a recent study showed that alpha emitters were more efficient in cancers with heterochromatin states [81]. Thus, proliferation arrest and chromatin condensation may increase DNA damage and slow down DNA repair mechanisms.

Exposure of the alpha radiation-sensitive DP.C to alpha particles after proliferation arrest non-significantly increased the level of cytotoxicity at the low level of radiation (0.7 Gy) but not at higher radiation levels. This may indicate that lack of proliferation and probably chromatin condensation is not a significant factor in alpha radiation-inflicted cell damage and death. We may conclude that condensation of the DNA due to serum starvation marginally increased the cytotoxic effect of alpha radiation. Furthermore, unlike in the case of photon radiation, proliferation arrest did not reduce the cytotoxic effect of alpha radiation on melanoma cell lines.

In this respect, other studies found that cells with maximal chromatin compactness are more resistant to gamma radiation, but the damage is poorly repaired [82,83,84].

More on this aspect was reported in another study which showed that chromatin opening by histone deacetylase inhibitor trichostatin A (TSA) pretreatment reduced clonogenic survival and increased H2AX foci in MDA-MB-231 cells, indicative of increased damage induction by free radicals using gamma radiation. In contrast, TSA pretreatment tended to improve survival after alpha radiation while H2AX foci were similar or lower [81].

### 4.6. DNA Damage Control and Alpha Radiation

Next, we looked for combinations of chemotherapeutic regents that may enhance the cytotoxic effect of alpha radiation on the cutaneous melanoma cell lines.

Combined treatment of radiotherapy and chemotherapeutic reagents for cancers has advanced significantly in the recent years. The combination was suggested to reduce the side effects of treatment, and to overcome radiation resistance, it was suggested to target DNA damage repair [85]. Berzosertib is an example of a reagent that can inhibit the ATR pathway, and it is a potent inhibitor of the ATM pathway as well, which are responsible for sensing DNA damage in dividing cells. The ATM is primarily involved in the response to DSB while the ATR involved in a wide range of DNA damage [86]. It has been reported that combining X-ray radiation and an ATR inhibitor (ATRi) enhanced the effect of radiation in non-small cell lung carcinoma (NSCLC) [87] and that combining ATRi with radiation is more effective than the treatments alone in head and neck squamous cell carcinoma [88]. Moreover, ATRi was found to radio-sensitize triple negative breast cancer cells [89]. Therefore, we tested whether ATR inhibition by berzosertib would enhance the cytotoxic effect of irradiated cutaneous melanoma cell lines. Cells were pretreated with berzosertib (100, 300, 500 nM) for 24 h, followed by increasing doses of alpha radiation, and compared to irradiated-only. Results showed that pretreatment with berzosertib 24 h before radiation marginally enhanced cytotoxicity in all four cutaneous cell lines. It should be mentioned that attempts to sensitize cells to alpha radiation by inhibitors of DNA damage repair (DDR) revealed that some cells can be sensitized by DNA-PK inhibitors while others by ATM inhibition [90].

Another chemotherapeutic reagent that was used in our study was the semi-synthetic compound CPT- 11 (irinotecan), which is an inhibitor of topoisomerase 1, and its potency in enhancing radiosensitivity has been known for almost thirty years. Irinotecan radio-sensitized human melanoma variant (U1-mel) to X-ray radiation [91], and the human breast cancer cell line MCF-7 was also radio-sensitized by irinotecan [92]. The combined treatment results of irinotecan and alpha radiation showed evidence for radio-sensitization by the drug to alpha radiation. Yet irinotecan was highly toxic to the melanoma cells, and a combined effect was observed with sublethal dose of 1 µM for 24 h pre radiation. Although more research is needed, we can conclude that irinotecan, as a modifier for the inhibition of the Topoisomerase 1, can be considered as a radio-sensitization agent for alpha radiation.

## 5. Conclusions

In our study, we investigated the cytotoxic effect of alpha radiation on human-derived melanoma cell lines. These cell lines were found to be sensitive to alpha radiation, and compared to gamma radiation, alpha radiation was found to be more lethal. Furthermore, we observed inter-tumor heterogeneity in sensitivity to alpha radiation, which might correlate with cell nuclear morphology and the amount of DSB. Furthermore, chromatin condensation, due to serum starvation, did not enhance the cytotoxic effect of alpha radiation. Topoisomerase 1 inhibitor was shown to be a potential radio-sensitization agent to the cytotoxic effect of alpha radiation while the ATM/ATR inhibitor (berzosertib) showed only marginal activity in combination with alpha radiation. Alpha radiation has a strong cytotoxic effect due to direct DNA damage and activation of death signals; nevertheless, death kinetics measurements demonstrated that it can cause proliferation arrest which may result in mitotic catastrophe. We may conclude that the inflicted damage of alpha radiation may kill the cell by a combination of mechanisms, such as destruction of vital organelles, irreparable DNA damage, and cell growth arrest.

Comparison of melanoma cells with an SCC cell line, CAL 27, indicated that nucleus size and number of DSB are not the only factors which govern the sensitivity to alpha radiation, and DNA damage configuration, repair mechanisms, and cell death signals may also figure importantly in this sensitivity.

These results may pave the way to treat human melanoma by alpha radiation-based treatments such as Alpha DaRT.

## Figures and Tables

**Figure 1 cancers-16-03804-f001:**
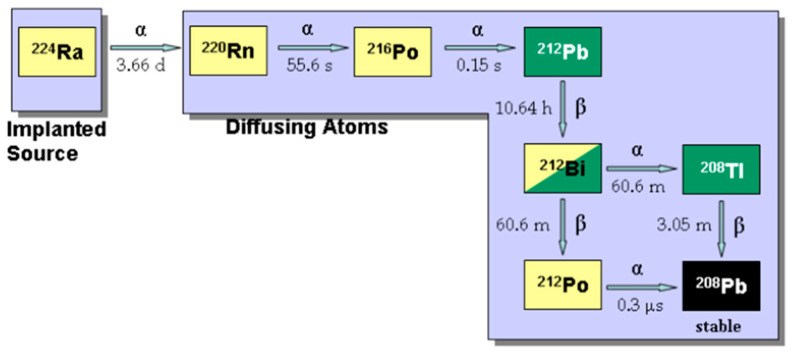
The Ra-224 decay chain.

**Figure 2 cancers-16-03804-f002:**
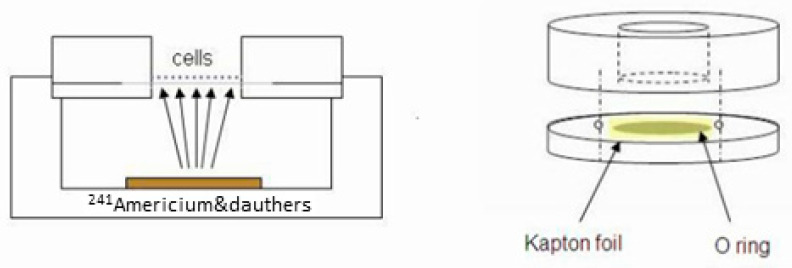
Schematic drawing of the alpha particle in vitro irradiation apparatus. The setup consists of two elements: A stainless-steel cylinder in which cells are seeded on a thin Kapton foil (irradiation cell holder) (**right**). The cell holder is positioned on top of the alpha particle irradiator composed of a well containing a sealed Am-241 source (**left**). Alpha particles emitted from the sealed Am-241 source pass through the Kapton foil, irradiating the seeded cells.

**Figure 3 cancers-16-03804-f003:**
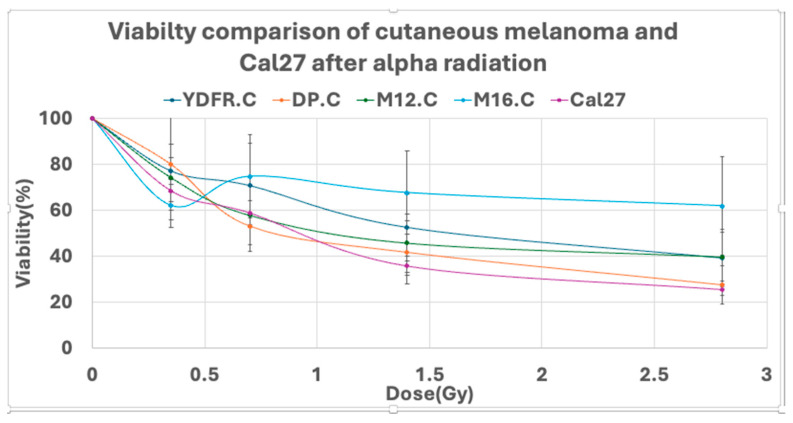
Comparison of the effect of alpha radiation on cutaneous melanoma cells and SCC CAL 27 cells. Viability curves at various radiation doses.

**Figure 4 cancers-16-03804-f004:**
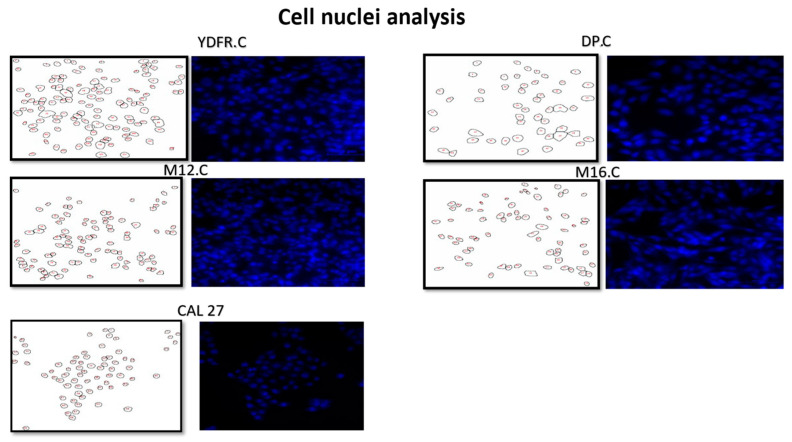
Cell lines nuclei analysis. Scale bar of 100 µm. Variants’ immunofluorescence photos dyed with DAPI (**right**), and outlines of cell nuclei after thresholding (**left**). Pixel ratio scale of 1–1.1. Candidate nuclei were normalized with the analyzer of particles of the software, with a circularity of 0.7–0.9.

**Figure 5 cancers-16-03804-f005:**
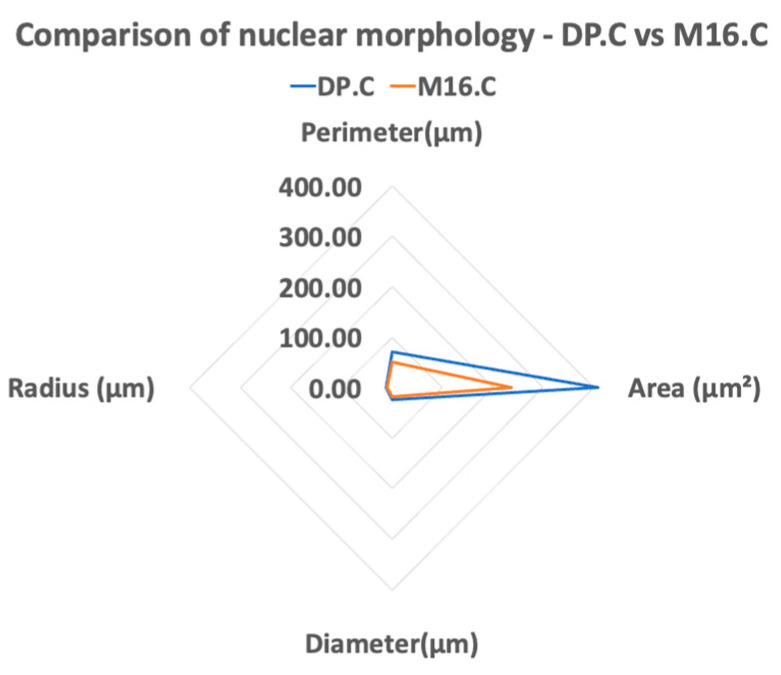
Multiparametric morphological measurements of cell lines’ nuclei. Comparison between the sensitive, DP.C, and resistant, M16.C, melanoma cell variants’ nuclei means: area [µm^2^], diameter [µm], perimeter [µm], radius of nuclei [µm].

**Figure 6 cancers-16-03804-f006:**
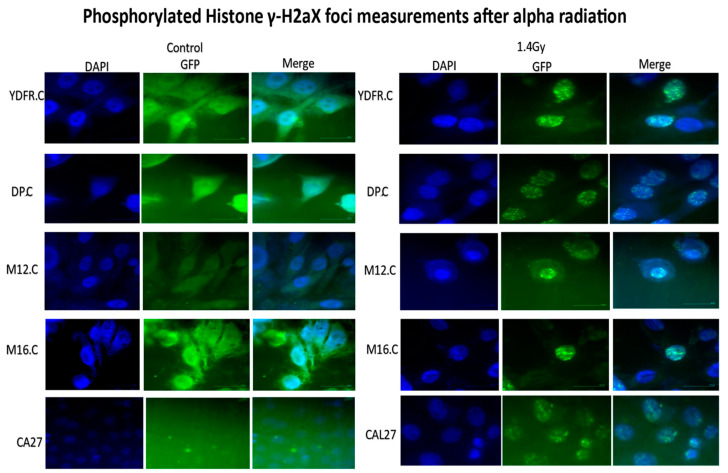
Phosphorylated foci of four cell variants. Immunofluorescence staining of the phosphorylated histone γ-H2AX after irradiation of cells with dose of 1.4 Gy of alpha radiation. Cell variants were irradiated with 1.4 Gy and immediately fixed with 4% PFA, perforated with 0.5% Triton-X, blocked with 10% FBS in PBS solution, and treated with primary and secondary antibodies. Cell nuclei were dyed with DAPI. Foci per nucleus were analyzed. Foci were counted manually, and an ANOVA test conducted with a confidence interval of 95% showed significant variance between our subjects (*p* = 1.06 × 10^−8^). Post hoc analysis of the results revealed significant variance among pairs of cell variants: DP.C:M12.C (*p* = 1 × 10^−8^), DP.C:M16.C (*p* = 5.64 × 10^−6^), DP.C:YDFR.C (*p* = 0.002), M12.C:YDFR.C (*p* = 0.037). Moreover, non-significant results were obtained in the analysis: M12.C:M16.C (*p* = 0.53) and M16.C:YDFR.C (*p* = 0.49).

**Figure 7 cancers-16-03804-f007:**
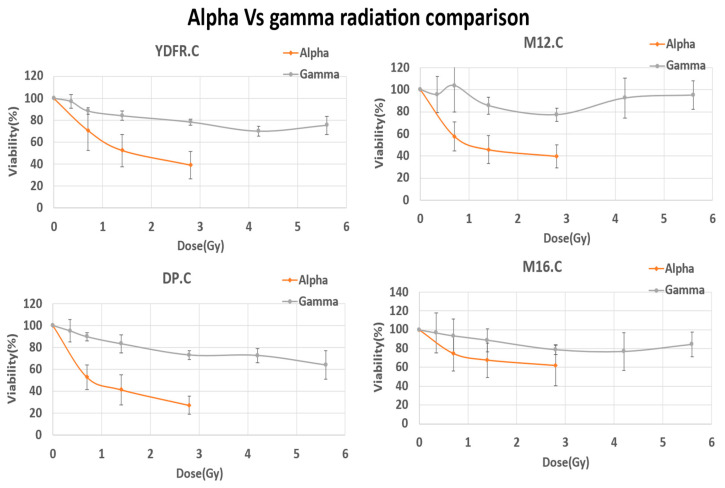
Effect of alpha and gamma radiation on melanoma cells. Four cell variants of human-derived melanoma were subjected to both alpha and gamma radiation with increasing doses. Percent viability was evaluated with colorimetric assays (HemaColor, N = 3 for each variant). Alpha radiation (orange lines) achieved higher cytotoxicity levels than gamma radiation (blue lines) at the same radiation doses. Paired *t*-test of the D50 measurements of alpha radiation compared to gamma suggests significance among all four cell variants (YDFR.C *p* = 0.004, DP.C *p* = 0.001, M12.C *p* = 0.025, M16.C *p* = 0.0063).

**Figure 8 cancers-16-03804-f008:**
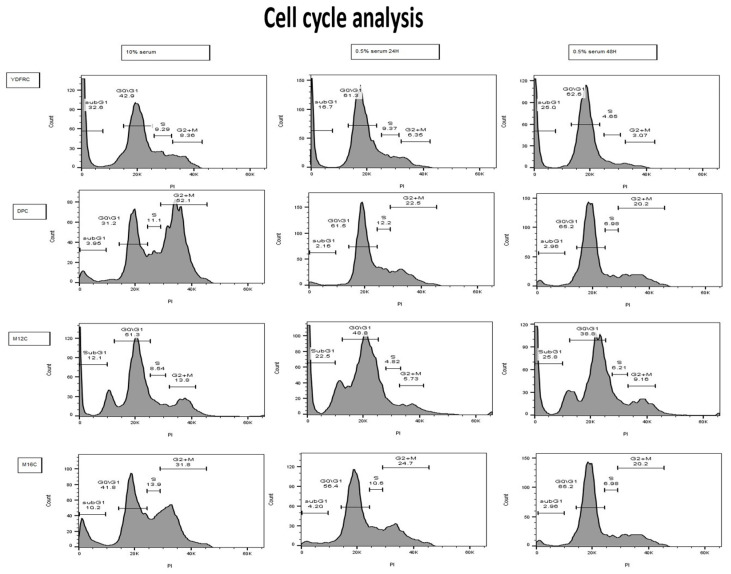
Cell cycle analysis. Cells labeled with PI were processed with 10% serum and 0.5% serum after 24 and 48 h of incubation. Starvation medium cells underwent cell cycle arrest in three of four cell lines. The M12.C cell line underwent an increase in the sub-G1 phase after starvation (N = 6). Three of the four cell lines underwent cell cycle arrest after approximately 24 h of incubation with starvation medium.

**Figure 9 cancers-16-03804-f009:**
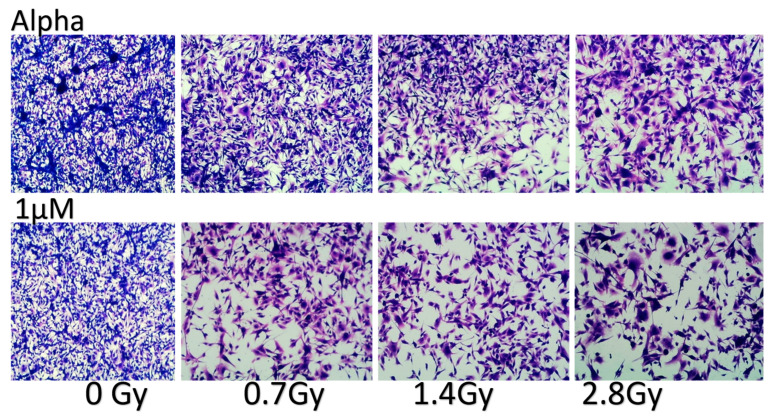
Photos of the combined treatment of alpha radiation with irinotecan. HemaColor-stained cells of the YDFR.C cell variant after the combined treatment with different concentrations of irinotecan (1, 3, and 5 µM) and alpha radiation.

**Figure 10 cancers-16-03804-f010:**
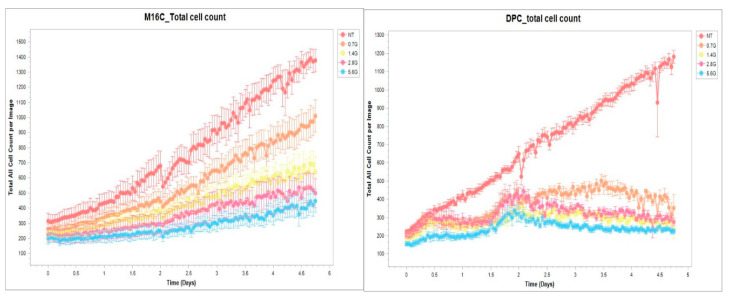
Total cells count after radiation. DP.C and M16.C cutaneous cells were examined in the IncuCyte machine for five days. Untreated cells were compared to irradiated cells with 0.7, 1.4, 2.8, and 5.6 Gy of alpha radiation. Total cell count was analyzed with the 2022B Rev2 analyzer.

**Table 1 cancers-16-03804-t001:** Analysis of alpha radiation effects on melanoma cell variants.

Alpha Radiation D50 Measurements ^a^			
Variant	N	D50 (Gy)	SD (Gy)
YDFR.C	7	2.05	0.64
YDFR.CB3	5	1.35	0.53
DP.C	6	1.51	0.35
DP.CB2	3	1.2	0.18
M12.C	7	1.87	0.54
M12.CB3	3	2.41	0.44
M16.C	6	2.64	0.74
M16.CB3	4	2.04	0.12
CAL 27	3	1.37	0.07

^a^ Four cutaneous cell lines and their brain metastatic derivatives were irradiated with ongoing doses of alpha radiation (0–2.8 Gy). Summary of the D50 measurement for 50% cell viability/cytotoxicity for each cell variant with the SD and the number of viability assay repetitions for each cell variant (N). The D50 was calculated with a linear line formula. ANOVA test was applied and showed significant variance (*p* = 0.0263) between the four cutaneous cell lines. Post hoc analysis showed significant variance between DP.C and M16.C (*p* = 0.0173) and no variance between the other cutaneous cell lines.

**Table 2 cancers-16-03804-t002:** Summarized measurements.

Cell Nuclei Morphology				Alpha Radiation D50 Measurements				γ-H2AX Foci			
Variant	N	Area (µm^2^)	SD (µm^2^)	Variant	N	D50 (Gy)	SD (Gy)	Variant	N	Foci Per Nuclei	SD
**DP.C**	50	408.83	19.71	**DP.C**	6	1.51	0.35	**DP.C**	33	9.6	2.29
**YDFR.C**	122	364.49	22.64	**YDFR.C**	7	2.05	0.64	**YDFR.C**	28	7.67	2.12
**M16.C**	70	237.00	10.98	**M16.C**	6	2.64	0.74	**M16.C**	31	6.9	1.65
**M12.C**	96	196.35	6.48	**M12.C**	7	1.87	0.54	**M12.C**	29	6.17	2.22
**CAL 27**	362	175.49	70.91	**CAL 27**	3	1.36	0.07	**CAL 27**	121	7.585	2.07

Summarized measurements of four cutaneous melanoma cell variants including the derived D50, nuclear perimeter, the calculated nuclear area according to the derived radius, and the mean number of phosphorylated foci per cell variant after alpha radiation exposure.

**Table 3 cancers-16-03804-t003:** Cell nuclei morphology of cutaneous cell lines.

Cell Nuclei Morphology ^a^									
Variant	N	Perimeter (µm)	SD (µm)	Radius (µm)	SD (µm)	Diameter (µm)	SD (µm)	Area (µm^2^)	SD (µm^2^)
**YDFR.C**	122	67.66	16.86	10.77	2.68	21.54	5.37	364.49	22.64
**DP.C**	50	71.65	15.73	11.41	2.50	22.82	5.01	408.82	19.70
**M12.C**	96	49.66	9.02	7.90	1.43	15.81	2.87	196.34	6.47
**M16.C**	70	51.41	13.18	8.68	1.87	17.37	3.74	236.99	10.98
**Cal 27**	362	46.09	9.01	7.34	1.43	14.67	2.86	175.49	70.91

^a^ Mean cell nuclear morphology parameters of the cutaneous melanoma variants and the comparison with CAL 27. ANOVA test of the extracted perimeter of cutaneous cell variants revealed strong significance (*p* = 2 × 10^−26^) and post HSD hoc revealed significance between pairs of cell variants YDFR.C and DP.C and M12.C and M16.C. Two tailed *t*-test revealed significance (*p* < 0.0001) between calculated nuclear area of cutaneous melanoma cell lines and CAL 27.

## Data Availability

All the data pertinent to this article are provided in the text.

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
