# Peer review of "Melanoma Cells from Different Patients Differ in Their Sensitivity to Alpha Radiation-Mediated Killing, Sensitivity Which Correlates with Cell Nuclei Area and Double Strand Breaks"

_cancers, 2024, doi:10.3390/cancers16223804_

Round 1

Reviewer 1 Report

Comments and Suggestions for Authors

I wish to congratulate the authors on their excellent work. They discuss a very important, clinically relevant topic, presenting the most relevant and recent data on alpha particle radiation and its utility in the treatment of various cancers, including melanoma. The authors critically interpret the published data and bring new perspectives regarding the cytotoxic effect of alpha particle radiation on melanoma and squamous cell carcinoma derived cell lines. Apart from a solid theoretic background, the article presents the results of the authors’ original study, the first of its kind. The manuscript is well organized, the study design is clear, the results are presented in a straightforward manner and the data analysis is done systematically.

I have a few suggestions that I believe the manuscript would benefit from:

1.     “…melanoma is more dangerous because of its ability to metastasize to other organs more rapidly if it is not treated early” (lines 52-53) Please consider commenting on the aggressive and unpredictable nature of melanoma.

2.     “EBRT therapy is an important treatment for cancer patients, with 50% of patients receiving radiation therapy during their illness, and it is considered to contribute to 40% of the cure of cancer [4].” (lines 58-60) Please refer to the utility of radiation therapy in melanoma, particularly in melanoma metastases.

3.     It is advisable to include a diagram illustrating the principle of Diffusing alpha-emitters Radiation Therapy (Alpha DaRT).

4.     “Indications of radiotherapy are currently reduced, since melanoma is traditionally considered relatively resistant to conventional photon radiotherapy [32,33]. (lines 117-118) Please comment on the use of radiation therapy in metastatic melanoma and as an adjuvant therapy.

5.     “Firstly, to see the response of human melanoma cells to alpha radiation, and secondly to learn about the variance in response within a group of cells from the same histotype, and thirdly to determine cell characteristics which might correlate with sensitivity/resistance of melanoma cells to alpha radiation.” (lines 124-126)  Please consider removing this paragraph as it is repetitive.

6.     “The radiobiological advantages of alpha particles include short range in tissue (typically 30-90 µm), high linear energy transfer (high-LET, 5,000-8,000 keV) and independence of the administered dose rate, oxygen effects and the cells state in the cell cycle during the irradiation [11]. (lines 610-613) Please consider removing this paragraph as it is repetitive.

7.     “In the current study, firstly, we measured the response of human melanoma cells to alpha radiation, secondly, studied the variance in response within a group of cells from  the same histotype, and thirdly looked for cell characteristics which might correlate with sensitivity/resistance of melanoma cells to alpha radiation. To the best of our knowledge, similar studies of this kind assessing the biological effect of alpha radiation on a battery  of human melanoma were not yet reported in the literature.” (lines 630-634) The objectives of the study and its novelty have already been stated in the introduction section.

8.     “These results are corroborated by previous studies in which treatment of cells with alpha  radiation or x-rays was compared.” (lines 646-647) Please add references.

9.     A diagram illustrating the ATM-initiated signaling cascade would make the data more accessible and understandable.

10.  The manuscript would benefit from a more detailed discussion on future perspectives based on the results of this study.

Best regards!

Author Response

Cancers reviewer 1

We thank the reviewer for the constructive criticism which will help to improve the paper

Comments and Suggestions for Authors

I wish to congratulate the authors on their excellent work. They discuss a very important, clinically relevant topic, presenting the most relevant and recent data on alpha particle radiation and its utility in the treatment of various cancers, including melanoma. The authors critically interpret the published data and bring new perspectives regarding the cytotoxic effect of alpha particle radiation on melanoma and squamous cell carcinoma derived cell lines. Apart from a solid theoretic background, the article presents the results of the authors’ original study, the first of its kind. The manuscript is well organized, the study design is clear, the results are presented in a straightforward manner and the data analysis is done systematically.

I have a few suggestions that I believe the manuscript would benefit from:

COMMENT 1.     “…melanoma is more dangerous because of its ability to metastasize to other organs more rapidly if it is not treated early” (lines 52-53) Please consider commenting on the aggressive and unpredictable nature of melanoma.

ANSWER: A paragraph was added.

COMMENT 2.     “EBRT therapy is an important treatment for cancer patients, with 50% of patients receiving radiation therapy during their illness, and it is considered to contribute to 40% of the cure of cancer [4].” (lines 58-60) Please refer to the utility of radiation therapy in melanoma, particularly in melanoma metastases.

ANSWER: ADDED, The role of adjuvant RT for high-risk resected melanoma remains limited [Keung EZ, Gershenwald JE. Clinicopathological Features, Staging, and Current Approaches to Treatment in High-Risk Resectable Melanoma. J Natl Cancer Inst. 2020 Sep 1;112(9):875-885. doi: 10.1093/jnci/djaa012].

COMMENT 3.     It is advisable to include a diagram illustrating the principle of Diffusing alpha-emitters Radiation Therapy (Alpha DaRT).

ANSWER: A diagram was added as recommended

COMMENT 4.     “Indications of radiotherapy are currently reduced, since melanoma is traditionally considered relatively resistant to conventional photon radiotherapy [32,33]. (lines 117-118) Please comment on the use of radiation therapy in metastatic melanoma and as an adjuvant therapy.

ANSWER: See answer to comment  2.

COMMENT 5.     “Firstly, to see the response of human melanoma cells to alpha radiation, and secondly to learn about the variance in response within a group of cells from the same histotype, and thirdly to determine cell characteristics which might correlate with sensitivity/resistance of melanoma cells to alpha radiation.” (lines 124-126)  Please consider removing this paragraph as it is repetitive.

ANSWER;  Removed as suggested

COMMENT 6.    “The radiobiological advantages of alpha particles include short range in tissue (typically 30-90 µm), high linear energy transfer (high-LET, 5,000-8,000 keV) and independence of the administered dose rate, oxygen effects and the cells state in the cell cycle during the irradiation [11]. (lines 610-613) Please consider removing this paragraph as it is repetitive.

ANSWER: Removed as suggested.

  1. “In the current study, firstly, we measured the response of human melanoma cells to alpha radiation, secondly, studied the variance in response within a group of cells from  the same histotype, and thirdly looked for cell characteristics which might correlate with sensitivity/resistance of melanoma cells to alpha radiation. To the best of our knowledge, similar studies of this kind assessing the biological effect of alpha radiation on a battery of human melanoma were not yet reported in the literature.” (lines 630-634) The objectives of the study and its novelty have already been stated in the introduction section.

ANSWER: These lines were removed as suggested.

COMMENT 8.     “These results are corroborated by previous studies in which treatment of cells with alpha  radiation or x-rays was compared.” (lines 646-647) Please add references.

ANSWER: the relevant references are  [Lazarov, Nishri]

COMMENT 9.     A diagram illustrating the ATM-initiated signalling cascade would make the data more accessible and understandable.

ANSWER: We did not add such a diagram because we think it may overload the paper.

COMMENT 10.  The manuscript would benefit from a more detailed discussion on future perspectives based on the results of this study.

ANSWER: A statement on the future use of alpha radiation against melanoma was added to the conclusions.

Reviewer 2 Report

Comments and Suggestions for Authors

The Authors present an important paper on alpha particle radiation of melanoma. The results might lead to new options for melanoma radiation therapy.

However, there are some major concerns regarding how the conclusions were formulated. The conclusions drawn from the data presented are not correct.

Major issues:

- in order to compare the biological effects of different types of radiation usually an equivalent dose in Sv should be used, instead of dose delivered in Gy? The coefficient of alpha particles is 20, so this may produce a totally different results in Fig 5. Please justify your approach. Another issue with Fig 5 is why using % of viability, instead of standard SF on a log scale?

- the results shown in Table 1 demonstrate minute differences in the response to alpha particles. Only some of the differences were significant. Even if some of them are significant, I am not convinced that the main conclusion is supported, i.e. that “Melanoma cells were more resistant to alpha radiation when compared with the response of another skin tumor, squamous cell carcinoma.” And you also state that they exhibit a “varied response” to alpha particle radiation. In my opinion the response (i.e. D50) is very similar, and the difference in D50 between DP.C and CAL cell lines in fact none. The differences are described as “non-significant, yet considerable” (line 347), this is a misleading way of interpreting the data. Have you tried to calculate alpha and beta parameters of the linear-quadratic model from your data? Also, it might helpful to discuss the heterogeneity in the melanoma cell responses to Xrays – what is the variance there and how much heterogeneity of response is to be expected for low-LET radiation (i.e. ref 54)? In Fig 5 when comparing different cell lines and two types of radiation – I agree that alpha particles are more effective, however there are no differences between the cell lines. The results lie within the SD of each other. You state in line 463: “It is interesting to note that the cell line most sensitive to alpha radiation (DP.C) was also more sensitive to photon radiation”. I do not agree, they are all very similar.

Minor comments:

- the alpha particle source had 0.25 Gy/ min. What was the dose rate for the gamma source and was it comparable? What was the LET of gamma source?

- while it is very commendable to state your aims clearly, in the manuscript they are a bit too much – first in lines 124-129, then repeated in 130-132, and then again a list of specific aims given in lines 134 – 150. Please unify and avoid repetition.

- varied font size throughout the manuscript

- sentence in line 246-247 is not necessary in p. 2.5

- why is 2.7 and 2.8 separate points?

- lines 307-316 should be in Discussion, and not in Methods. The description of the synchronizing the cells and serum starvation is missing.

- lines 319-323 are also part of discussion.

- please mark the significant differences directly in the Table 1.

- In Fig 2 caption, the sentence “Pixel ratio scale of 1-1.1. candidate nuclei were normalized with the analyzer of particles of the software with circularity of 0.7-0.9.” is unclear. This should be explained more in the Methods.

- why only 2 cell lines were shown in Fig 3 and Table 3? Could a correlation for all cell lines between some parameter of morphology of nucleus and radiosensitivity to alpha particles be performed? Or if not, al least three cell lines you mention – DP.C, M16.C and CAL?

- you do not have to repeat the methods in the results section, eg. line 456-7

- lines 504-519 are methods, not results

- lines 534-542 are methods, not results and so on in the following parts of the manuscript p 3.9.2, 3.10, etc.

- Fig 7 is not quantitative, so it should be put in the supplement.

-              To fully demonstrated the advantage, it would be great to see the effect of alpha particle irradiation in hypoxic vs normoxic conditions, including the efficiency of the DNA repair mechanisms in different cell lines – perhaps for the next paper

-              Line 624: lines were shown to exhibit the heterogeneity in which biological feature?

-              Instead of the phrase “battery of melanoma cells” usually “a panel of melanoma cells” is used

-              Please add some discussion on DNA repair mechanisms differences between gamma and alpha radiation. Are there any mutations in melanoma that may interfere with DNA repair?

-              Line 704 you mention clustered DNA damage – has any clustered DNA damage sites been spotted in your results?

-              Lines 743-750: do you think these differences might be related to the efficiency of the DNA repair mechanisms in these cells? Are there any literature data on this?

-              Line 770: “increased non significantly the level of cytotoxicity” this is again, incorrect statement. If the effect is not statistically significant, that means it is within the noise, i.e. there is no effect!!

-              the Discussion is very long – it would be advisable to divide it into smaller sections, each with a title describing its content and the conclusion it concerns

-              the conclusion that alpha particles are more lethal is hardly new

-              “death kinetic measurements” – if you mean results in Fig 8, these are cell growth kinetics, not death kinetics

Author Response

We thank the reviewer for his thorough analysis of the manuscript and important suggestions and corrections.

Comments and Suggestions for Authors

The Authors present an important paper on alpha particle radiation of melanoma. The results might lead to new options for melanoma radiation therapy.

However, there are some major concerns regarding how the conclusions were formulated. The conclusions drawn from the data presented are not correct.

Major issues:

COMMENT:  in order to compare the biological effects of different types of radiation usually an equivalent dose in Sv should be used, instead of dose delivered in Gy? The coefficient of alpha particles is 20, so this may produce a totally different results in Fig 5. Please justify your approach.

ANSWER: The biological coefficient of 20 for alpha particles is valid in the context of radiological damage to healthy tissue. For the therapeutic effect on tumors, we expect a completely different factor. In fact, one of the reasons of performing comparative studies is to obtain this factor.  Quite generally (based also on previous research of our group) the factor is in the range of 3 to 5, depending on tumor type.

COMMENT: Another issue with Fig 5 is why using % of viability, instead of standard SF on a log scale?

ANSWER:  These melanoma cells do not form colonies in tissue culture that can be counted after irradiation. They rather grow in dispersion which requires to measure the total cells in culture, which was done by the Hemacolor staining method.

COMMENT:  the results shown in Table 1 demonstrate minute differences in the response to alpha particles. Only some of the differences were significant. Even if some of them are significant, I am not convinced that the main conclusion is supported, i.e. that “Melanoma cells were more resistant to alpha radiation when compared with the response of another skin tumor, squamous cell carcinoma.” And you also state that they exhibit a “varied response” to alpha particle radiation.

ANSWER: Many thanks to the reviewer who pointed out a mistake in the manuscript. The legend to table 1 refers to an old version of the paper. More experiments were performed, and the updated statistical analysis is now presented.

New legend to Table 1.  Analysis of alpha radiation effect on cell lines of human melanoma. Four cutaneous cell lines and their brain metastatic derivatives were irradiated with ongoing doses of alpha radiation (0-2.8 Gy). Summary of the D50 measurement for 50% cell viability/Cytotoxicity for each cell variant with the SD and the number of viability assay repetitions for each cell variant(N). The D50 were calculated with linear line formula. ANOVA test was applied and showed significant variance (P=0.0263) between the four cutaneous cell lines. Post–Hoc analysis showed significant variance between DP.C to M16.C (P=0.0173) and no variance between the other cutaneous cell lines.

COMMENT: In my opinion the response (i.e. D50) is very similar, and the difference in D50 between DP.C and CAL cell lines in fact none. The differences are described as “non-significant, yet considerable” (line 347), this is a misleading way of interpreting the data.

ANSWER: The comparison between melanoma and was recalculated and the results presented in section 3.2. “CAL 27 (D50=1.37+/-0.07 Gy), was more sensitive than all melanoma cells but paired T-test analysis which was conducted on CAL27 and the cutaneous cell lines revealed significant variance between CAL27 and the most resistant cell, M16.C (P=0.0264)”.  We also added a new figure to demonstrate the different response of melanoma cells and SCC cells to alpha radiation (Figure 2).

COMMENT: Have you tried to calculate alpha and beta parameters of the linear-quadratic model from your data?

ANSWER: The parametrization in terms of the linear-quadratic model is out of the scope of the present research.

COMMENT: Also, it might be helpful to discuss the heterogeneity in the melanoma cell responses to X-rays – what is the variance there and how much heterogeneity of response is to be expected for low-LET radiation (i.e. ref 54)?

ANSWER: The focus of this paper was the response of melanoma cells to alpha radiation and to find out if alpha radiation is more effective than photon radiation in killing melanoma cells. Therefore, we did not analyze the response of melanoma cells to photon radiation.

COMMENT: In Fig 5 when comparing different cell lines and two types of radiation – I agree that alpha particles are more effective, however there are no differences between the cell lines. The results lie within the SD of each other. You state in line 463: “It is interesting to note that the cell line most sensitive to alpha radiation (DP.C) was also more sensitive to photon radiation”. I do not agree, they are all very similar.

ANSWER: since we did not analyze the response of the cells to photon radiation we remove this statement from the paper.

Minor comments:

COMMENT:  the alpha particle source had 0.25 Gy/ min. What was the dose rate for the gamma source and was it comparable? What was the LET of gamma source?

ANSWER: According to the manufacturer the dose rate of the gamma source stands for 2.5-5 Gy\min depending on the model of the machine. The radiation times were adjusted to be equivalent to the doses of alpha.

The LET for gamma sources depends on the energy of the radiation, but it is generally about two orders of magnitude smaller than for charged particles. Note that there is a basic difference in the physical mechanism of interaction of the two species with matter.

COMMENT: while it is very commendable to state your aims clearly, in the manuscript they are a bit too much – first in lines 124-129, then repeated in 130-132, and then again, a list of specific aims given in lines 134 – 150. Please unify and avoid repetition. Varied font size throughout the manuscript

ANSWER: corrected

COMMENT: sentence in line 246-247 is not necessary in p. 2.5 

COMMENT: why is 2.7 and 2.8 separate points?

ANSWER: the two pointes were combined

COMMENT: lines 307-316 should be in Discussion, and not in Methods.

ANSWER: Moved to discussion

COMMENT: The description of the synchronizing the cells and serum starvation is missing.

ANSWER: Added in section 2.9

COMMENT:  lines 319-323 are also part of discussion.

ANSWER: This is an introduction to the results section.

COMMENT: please mark the significant differences directly in the Table 1.

ANSWER: The significant differences are detailed in the footnote for table 1

COMMENT: In Fig 2 caption, the sentence “Pixel ratio scale of 1-1.1. candidate nuclei were normalized with the analyzer of particles of the software with circularity of 0.7-0.9.” is unclear. This should be explained more in the Methods.

ANSWER: This is not crucial for understanding the results and was removed.

COMMENT: why only 2 cell lines were shown in Fig 3 and Table 3? Could a correlation for all cell lines between some parameter of morphology of nucleus and radiosensitivity to alpha particles be performed? Or if not, at least three cell lines you mention – DP.C, M16.C and CAL.

ANSWER: We performed the analysis for all cell lines but present a comparison only for DP.C (the most sensitive cell) and M16.C (the most resistant cell).

COMMENT: you do not have to repeat the methods in the results section, eg. line 456-7:

ANSWER: Method description was removed from the results.

COMMENT:  lines 504-519 are methods, not results.

ANSWER: The description of the method was removed. Method is detailed in section 2.9.

COMMENT: lines 534-542 are methods, not results and so on in the following parts of the manuscript p 3.9.2, 3.10, etc.

ANSWER: Methods were moved to the method sections 2.10 and 2.11.

- Fig 7 is not quantitative, so it should be put in the supplement.

COMMENT:   To fully demonstrated the advantage, it would be great to see the effect of alpha particle irradiation in hypoxic vs normoxic conditions, including the efficiency of the DNA repair mechanisms in different cell lines – perhaps for the next paper J

ANSWER: This is indeed an important issue which is a whole project by itself and out of the scope of this study.

COMMENT:  Line 624: lines were shown to exhibit the heterogeneity in which biological feature?

ANSWER: See ref 52 Moshe, A.; et al. Inter-Tumor Heterogeneity-Melanomas Respond Differently to GM-CSF-Mediated Activation. Cells 2020, 9, 1683.

COMMENT:         Instead of the phrase “battery of melanoma cells” usually “a panel of melanoma cells” is used-

ANSWER: DONE

COMMENT:   Please add some discussion on DNA repair mechanisms differences between gamma and alpha radiation. Are there any mutations in melanoma that may interfere with DNA repair?

ANSWER: A paragraph was added in section 4.2.

COMMENT:    Line 704 you mention clustered DNA damage – has any clustered DNA damage sites been spotted in your results?

ANSWER: No.

COMMENT:   Lines 743-750: do you think these differences might be related to the efficiency of the DNA repair mechanisms in these cells? Are there any literature data on this?

ANSWER: We do assume that DNA repair mechanisms might be involved in the sensitivity to alpha radiation, but we do not have evidence for that. This is in our future plans. In another study we performed on mouse tumor cells of different histotypes we found that the speed of DNA repair is affecting survival after alpha radiation inflicted damage.

COMMENT:    Line 770: “increased non significantly the level of cytotoxicity” this is again, incorrect statement. If the effect is not statistically significant, that means it is within the noise, i.e. there is no effect!!

ANSWER: The statement is correct. An increase was observed and measured, and this is what we reported with the statistical analysis.

COMMENT:  the Discussion is very long – it would be advisable to divide it into smaller sections, each with a title describing its content and the conclusion it concerns

ANSWER: Discussion was divided into sections.

-              the conclusion that alpha particles are more lethal is hardly new

COMMENT:    “death kinetic measurements” – if you mean results in Fig 8, these are cell growth kinetics, not death kinetics-

ANSWER: Corrected.

Round 2

Reviewer 2 Report

Comments and Suggestions for Authors

The Authors corrected most of the mistakes and adapted to most of the comments.

However, I was puzzled by some of the answers to my comments, eg:

COMMENT: Also, it might be helpful to discuss the heterogeneity in the melanoma cell responses to X-rays – what is the variance there and how much heterogeneity of response is to be expected for low-LET radiation (i.e. ref 54)?

ANSWER: The focus of this paper was the response of melanoma cells to alpha radiation and to find out if alpha radiation is more effective than photon radiation in killing melanoma cells. Therefore, we did not analyze the response of melanoma cells to photon radiation.

Calculating variance of the cell response to gamma radiation from the data you already have is very simple. Why not analyze it?

The very next comment is totally bizzare, as it directly opposes the above:

COMMENT: In Fig 5 when comparing different cell lines and two types of radiation – I agree that alpha particles are more effective, however there are no differences between the cell lines. The results lie within the SD of each other. You state in line 463: “It is interesting to note that the cell line most sensitive to alpha radiation (DP.C) was also more sensitive to photon radiation”. I do not agree, they are all very similar.

ANSWER: since we did not analyze the response of the cells to photon radiation we remove this statement from the paper.

You most certainly have the response of cells to photon, i.e. gamma radiation!

In the following comment and answer the main question was - do you have more data to perform a correlation? And it seems you have, yet again you avoid presenting it more fully.

COMMENT: why only 2 cell lines were shown in Fig 3 and Table 3? Could a correlation for all cell lines between some parameter of morphology of nucleus and radiosensitivity to alpha particles be performed? Or if not, at least three cell lines you mention – DP.C, M16.C and CAL.

ANSWER: We performed the analysis for all cell lines but present a comparison only for DP.C (the most sensitive cell) and M16.C (the most resistant cell).